# Dietary Supplementation with Yerba Mate (*Ilex paraguariensis*) Infusion Increases IRS-1 and PI3K mRNA Levels and Enhances Insulin Sensitivity and Secretion in Rat Pancreatic Islets

**DOI:** 10.3390/plants12142620

**Published:** 2023-07-12

**Authors:** Bárbara Maiztegui, Hernán Gonzalo Villagarcía, Carolina Lisi Román, Luis Emilio Flores, José María Prieto, María Cecilia Castro, María Laura Massa, Guillermo R. Schinella, Flavio Francini

**Affiliations:** 1CENEXA, Centro de Endocrinología Experimental y Aplicada (UNLP-CONICET La Plata), La Plata 1900, Argentina; bmaiztegui@cenexa.org (B.M.); hvillagarcia@med.unlp.edu.ar (H.G.V.); clroman@cenexa.org (C.L.R.); leflores@cenexa.org (L.E.F.); mccastro@cenexa.org (M.C.C.); mlmassa@cenexa.org (M.L.M.); 2Facultad de Ciencias Médicas, Universidad Nacional de la Plata, La Plata 1900, Argentina; schinell@med.unlp.edu.ar; 3Centre for Natural Products Discovery, School of Pharmacy and Biomolecular Sciences, Liverpool John Moores University, Byrom Street, Liverpool L3 3AF, UK; j.prieto@ucl.ac.uk; 4Instituto de Ciencias de la Salud, UNAJ-CICPBA, Florencio Varela 1888, Argentina

**Keywords:** Yerba mate, Ilex paraguariensis, insulin secretion, insulin sensitivity, diabetes

## Abstract

“Yerba mate” (YM), an aqueous extract of Ilex paraguariensis, has antioxidant, diuretic, cardio-protective and hypoglycaemic properties. Since its effect on the pancreatic islets remains unclear, we evaluated insulin sensitivity and glucose-stimulated insulin secretion (GSIS) in rats consuming YM or tap water (C) for 21 days. Glucose tolerance, glycemia, triglyceridemia, insulinemia, TBARS and FRAP serum levels were evaluated. GSIS and mRNA levels of insulin signaling pathway and inflammatory markers were measured in isolated pancreatic islets from both groups. In C rats, islets were incubated with YM extract or its phenolic components to measure GSIS. YM improved glucose tolerance, enhanced GSIS, increased FRAP plasma levels and islet mRNA levels of IRS-1 and PI3K (p110), and decreased TBARS plasma levels and islet gene expression of TNF-α and PAI-1. Islets from C rats incubated with 100 µg/mL dry YM extract, 1 µM chlorogenic acid, 0.1 and 1 µM rutin, 1 µM caffeic acid or 1 µM quercetin showed an increase in GSIS. Our results suggest that YM enhances glucose tolerance because of its positive effects on GSIS, oxidative stress rate and insulin sensitivity in rat islets, suggesting that long-term dietary supplementation with YM may improve glucose homeostasis in pre-diabetes or type 2 diabetes.

## 1. Introduction

“Yerba mate” or “tea from Paraguay” (South American tea), commonly known as “Mate”, is a caffeine-containing, psycho-stimulant drink obtained from the leaves of the South American shrub *Ilex paraguariensis* A.St.-Hil. (Aquifoliaceae). This socializing beverage is widely consumed in Argentina, Uruguay, Paraguay, and some regions in Brazil. It has been introduced to the European and North American markets as both an herbal beverage and food supplement or herbal remedy with psycho-stimulant, antioxidant, diuretic, and tonic claims. In fact, Yerba mate has been amply shown to be enriched in various compounds with significant in vitro and in vivo biological activities, including antioxidant and anti-inflammatory properties [1,2,3,4]. Beyond its stimulating action on the central nervous system due to it containing caffeine and other xanthine alkaloids, Yerba mate also has positive vascular (vasodilation) and metabolic (lipid-lowering, hypoglycemic and weight reduction) effects largely attributed to the presence of phenolic components with high antioxidant capacity [5,6,7,8]. However, the underlying mechanisms of the anti-diabetic effect of Yerba mate are not fully understood, making it a subject of increasing interest in the research community. Yerba mate supplementation was reported to inhibit sodium–glucose transporter-1 (SGLT-1) expression in alloxan-induced diabetic rats [9]. Other studies have shown that it induces a reduction in food intake, lipid metabolism, and glycemia when administered to mice [10], improving insulin resistance in a murine model of high-fat diet [8,11,12]. In this model, administration of Yerba mate preparations completely reverse hepatic lipogenesis, its methylxanthines and polyphenols being responsible for lowering plasma levels of cholesterol, LDL, and triacylglycerols, while saponins enhance lipogenesis in adipose tissue and total fecal fat excretion [13]. It has also been demonstrated that saponins have an insulinotropic effect on MIN-6 and INS-1 beta-cell lines [14,15].

In addition, the phenolic compounds found in Yerba mate significantly improve the oral glucose tolerance curve and reduce protein glycation with both glucose (50%) and fructose (90%) [16].

Arcari et al. have demonstrated that Yerba mate increases liver and muscle levels of insulin receptor substrate-1 (IRS-1) as well as phosphorylation of protein kinase B (Akt), while also reduces NF-kB activation with the consequent reduction of TNF-α, IL-6 and iNOS gene expression [11].

Several studies have demonstrated the positive influence of Yerba mate consumption on people with pre-diabetes or type 2 diabetes (T2D). T2D is a serious invalidating chronic disease, expanding worldwide, characterized by an early and progressive loss of pancreatic β-cell mass together with a damaged function [17]. The endocrine pancreas is represented by the islets of Langerhans, which are distributed among exocrine tissue and constitute 2% of the total pancreatic mass [18]. It is well documented that pancreatic islets play a crucial role in maintaining glucose homeostasis. They secrete many peptide hormones such as glucagon from α-cells, insulin from β-cells, somatostatin from δ-cells and pancreatic polypeptide (PP) from PP-cells. In addition, several non-classical islet peptides with similar important effects on islet function have also been identified [19]. Insulin is the main modulator of glycemic homeostasis, and its secretion is tightly regulated in response to physiological demands under strict endocrine, paracrine, and neuronal control [18].

Administration of Yerba mate for 60 days, with or without nutritional intervention, caused a decrease in both fasting glucose and HbA1c together with an improvement of the lipid profile in people with T2D [20]. In addition, Yerba mate consumption combined with nutritional intervention was highly effective in decreasing serum lipid parameters in pre-diabetic individuals [20]. Boaventura et al. demonstrated that consumption of 1 L of Yerba mate infusion per day, independently of the dietary intervention, increased plasma, and blood antioxidant protection in patients with dyslipidaemia [21]. Consumption of Yerba mate also significantly reduced fasting blood glucose and HbA1c after 40 and 60 days in patients with pre-diabetes and T2D respectively [22].

As seen above, the hypoglycemic effects of Yerba mate in human patients have been thoroughly documented. However, and rather surprisingly, information is also missing regarding its effects on pancreatic islets, a key player in glycemic regulation as mentioned, and its possible contribution to the mechanism behind these protective effects. Therefore, the aim of our study was to evaluate, in normal rats, the potential effects of Yerba mate on insulin sensitivity and secretion in pancreatic islets.

## 2. Results

### 2.1. Composition of the Aqueous Extract of Yerba Mate

HPLC-DAD analyses (Figure 1) revealed the presence of chlorogenic acid (peak 4); caffeine (peak 5); rutin (peak 7) + 3,5-dicaffeoylquininic acid (peak 8, unresolved); 4,5-dicaffeoylquininic acid (peak 10). Peaks 1, 2, and 3 were not identified, but have the same UV profile as the other caffeoylquinic acids identified, which in similar HPLC conditions are reported to be a mixture of this class of compounds [23].

The concentration of the plant metabolites of interest in the water extract (before the addition of alcohol as preservative) was calculated as: chlorogenic acid 1.952 mg/mL (R2 = 0.993), caffeine 1.836 mg/mL (R2 = 0.999), caffeic acid 0.092 mg/mL (R2 = 1.000), rutin 0.192 mg/mL (R2 = 0.997), and quercetin 0.044 mg/mL (R2 = 0.999).

### 2.2. Body Weight, Food Intake, and Plasma Parameters

Animals from the YM group drank a significantly higher volume of Yerba mate infusion than C rats, which consumed tap water (YM: 30.58 ± 1.04 vs. C: 25.40 ± 0.37 mL/rat/day, *p* < 0.05). However, both groups of rats ate a similar amount of solid food (YM vs. C: 20.87 ± 0.41 vs. 20.06 ± 0.41 g/rat/day). Overall, their caloric intake was comparable (YM: 60.28 ± 1.20; C: 57.95 ± 1.18 Kcal/rat/day). Body weight gain (YM vs. C: 4.65 ± 0.28 vs. 4.71 ± 0.20 g/rat/day), plasma triglyceride, glucose, and insulin levels were similar in both groups. Therefore, consumption of Yerba mate aqueous extract did not significantly modify either HOMA-IR or HOMA-β values (Table 1).

However, YM animals showed a significant decrease in circulating levels of TBARS (*p* < 0.05), a marker of peripheral oxidative stress, together with an increase in FRAP levels (*p* < 0.05) (Table 1), demonstrating that Yerba mate aqueous extract was able to simultaneously decrease oxidative damage and increase antioxidant capacity.

### 2.3. Glucose Tolerance Test

Plasma glucose values measured 15, 30 and 60 min after glucose load were significantly lower in the YM group compared to C (Figure 2A). Consequently, the area under the glucose curve (AUC) during GTT was significantly lower in YM animals (Figure 2B).

### 2.4. Glucose Stimulated Insulin Secretion

Fresh islets of Langerhans obtained from rats in each group shown an enhanced glucose-stimulated insulin secretion as a function of glucose amount in the media (Figure 3A). While no differences were found among experimental groups at 3.3 mM glucose, islets from YM rats released larger amounts of insulin in response to 16.7 mM glucose compared to control animals (*p* < 0.05 vs. C; Figure 3A).

### 2.5. Islet mRNA Levels of Intracellular Signaling Proteins of Insulin Pathway

Islets isolated from YM rats showed a significant increase of IRS-1 and PI3K (p110 catalytic subunit) mRNA levels (*p* < 0.05; Figure 3C,E). However, no significant differences in insulin receptor and IRS-2 mRNA levels were found among groups (Figure 3B,D).

### 2.6. Direct Effect of Yerba Mate and Its Phenolic Components on Ex-Vivo Insulin Secretion of Untreated Pancreatic Islets

Figure 4 shows that insulin was secreted by isolated pancreatic islets in response to glucose in a dose-dependent manner in all conditions tested. At basal glucose concentration (3.3 mM) islets incubated with either Yerba mate freeze-dried extract, or the different phenolic compounds, released a similar amount of insulin compared to the control group. However, at high glucose concentration (16.7 mM), the presence in the incubation medium of 100 µg/mL Yerba mate freeze-dried extract, 1 µM chlorogenic acid, 0.1 and 1 µM rutin, 1 µM caffeic acid or 1 µM quercetin, significantly enhanced insulin secretion with respect to control conditions (*p* < 0.05) (Figure 4).

### 2.7. Gene Expression (mRNA Levels) of Inflammatory Mediators in Pancreatic Islets

The mRNA levels of inflammatory response factors (TNF-α and PAI-1) were significantly lower in islets from YM animals compared to those isolated from C rats (YM vs. C: TNF-α 0.25 ± 0.03 vs. 1 ± 0.02 and PAI-1 0.48 ± 0.09 vs.1 ± 0.03 relative arbitrary units; *p* < 0.05 vs. C), suggesting the induction of an anti-inflammatory effect at the pancreatic islet level by the Yerba mate freeze-dried extract.

## 3. Discussion

Our current results show that a three-week-long dietary intervention consisting in supplementation of water with 2% Yerba mate infusion (ad libitum) to normal rats improved glucose tolerance by reducing serum glucose during the glucose tolerance test and, consequently, lowering the area under the glucose curve. In agreement with these changes, islets isolated from Yerba mate-fed rats not only enhanced insulin secretion in response to glucose but also increased mRNA levels of key proteins involved in the insulin signaling pathway (IRS-1 and PI3-K). In addition, some individual phenolic compounds present in Yerba mate freeze-dried extract enhanced insulin secretion ex vivo at stimulating glucose concentration (16.7 mM glucose) compared to control animals. Our results also demonstrated that consumption of Yerba mate (2% *w*/*v*) did not modify plasma levels of glucose, triglycerides, or insulin, while it was able to decrease oxidative damage and increase antioxidant capacity.

In a previous report, our group demonstrated that insulin has a physiological autocrine stimulatory role in glucose metabolism and glucose-induced insulin secretion in the islets of Langerhans [24]. Therefore, the increased gene expression of key proteins of the insulin signaling pathway mediators in islets isolated from rats drinking 2% Yerba mate suggests that these animals became more sensitive to autocrine regulation. Even when Yerba mate effect on the PI3K-AKT pathway has been described to improve hepatic insulin signaling [11], our results demonstrate for the first time that this pathway is also modified within pancreatic islets. Therefore, we may assume that Yerba mate consumption increases the secretory function of pancreatic β-cells as well as their insulin sensitivity in experimental animals. In addition, all the phenolic compounds of Yerba mate (2% *w*/*v*) assayed in the current experiments compared to their controls, induced an enhancement in insulin secretion in isolated islets from untreated rats when incubated at high glucose concentration, thereby suggesting their contribution to the effect of this herbal preparation.

Our experiments show a significant decrease in insular mRNA expression of proinflammatory cytokines (TNFα and PAI-1). In a previous in vitro work, it has been shown that saponins and quercetin—both compounds present in Yerba mate—synergistically inhibit iNOS and COX-2 in activated macrophages through NFκB pathways [25]. In our study, short-term (three weeks) dietary exposure to Yerba mate infusions also induced a significantly reduced expression in TNF-α. These results underline the key role of natural products found in certain foods (in our case, Yerba mate infusion) in regulating the inflammation process.

It was previously reported that TNF-α interferes with the insulin receptor signaling pathway and with metabolism of glucose transporters, and in consequence may be linked to the etiology of type-2 diabetes. Its presence in the incubation media of INS-1 (a well-established pancreatic β-cell line) decreases glucose-induced insulin secretion without affecting total insulin content [26]. Moreover, TNF-α positively regulates PAI-1 gene expression which, in turn, induces insulin resistance, and metabolic abnormalities in liver, muscle, and fat during proinflammatory processes [27]. All together, these results suggest that TNF-α acts as a link between insulin resistance and enhanced PAI-1 in obesity [28].

Whether TNF-α can act locally, systemically or at both levels, strong evidence clearly establishes that TNF-α overexpression plays a role in the pathophysiology of insulin resistance [29].

Several in vivo studies have demonstrated the therapeutic potential of Ilex paraguariensis as an anti-inflammatory agent [4,8,30,31,32]. It was recently demonstrated that crude extract fractions, and the major compounds in Ilex paraguariensis, namely caffeine, rutin, and chlorogenic acid, showed anti-inflammatory effects in a murine model of pleurisy [33]. Moreover, some reports have demonstrated an enhanced insulin secretion in insulin-producing cell lines (Ins-1 β cell line) but not in pancreatic islets; when incubated with quercetin [34,35] and caffeic acid [36,37], they demonstrated an insulinotropic effect of rutin on rat isolated islets, an effect also recorded in our current experiments. Additionally, we demonstrate for the first time a direct effect of chlorogenic acid, caffeic acid, and quercetin on insulin secretion by pancreatic isolated islets. Thus, their insulinotropic action plus their previously known therapeutic actions as anti-inflammatory agents [38] suggests the contribution to these same activities exerted by the consumption of Yerba mate.

In conclusion, the current study demonstrates for the first time that short-term dietary exposition of normal Wistar rats to 2% Yerba mate infusion improved glucose tolerance. This effect seems to be related to higher insulin sensitivity of pancreatic islets, evidenced by both enhanced insulin secretion and activation of insulin signaling in Yerba mate-treated rats, mainly through the PI3K pathway. In vitro results demonstrated that the above-mentioned actions are probably mediated by some of the phenolic compounds present in Yerba mate. Enhanced sensitivity and glucose-induced insulin secretion are accompanied by an inhibition of proinflammatory marker expression at islet level, a decrease in circulating markers of oxidative damage, and a concomitant increase in plasma antioxidant capacity.

Taken together, these findings suggest that long-term dietary supplementation with Yerba mate may be beneficial for improving diseases characterized by glucose homeostasis alterations such as pre-diabetes or type 2 diabetes.

## 4. Materials and Methods

### 4.1. Chemicals and Drugs

Collagenase used for pancreatic islet isolation and FastStart SYBR Green Master mix used in qPCR reactions were provided by Roche Diagnostics GmbH (Mannheim, Germany). Bovine serum albumin (BSA, fraction V) was from Sigma Chemical Co. (St. Louis, MO, USA). DNase I and SuperScript III were provided by Gibco (Gibco-BRL, Waltham, MA, USA).

### 4.2. Plant Material and Extraction

Yerba mate was purchased from Taragüi (Las Marías, Corrientes, Argentina). This commercial product has been approved and certified by the Instituto Nacional de la Yerba Mate (Posadas, Argentina). Yerba mate was freshly prepared every day as an infusion of 20 g of Ilex paraguariensis A. St.-Hil. (Aquifoliaceae) dry, coarsely cut leaves added to 1 L of boiling tap water, stirred, and allowed to cool to 40 °C and then filtered through filter paper (Whatman, Millipore).

### 4.3. Experimental Animals

Normal male Wistar rats (230–260 g body weight) were maintained in a temperature-controlled facility (23 ± 2 °C, 50% humidity, and 12-h light–dark cycle, 06:00–18:00 h) and were randomly divided into two experimental groups: the control group (C) had ad-libitum access to a standard commercial diet and tap water, whereas the treatment group received the same diet and 2% Yerba mate (YM) for three weeks. We used 18 animals per group divided into three independent experiments with six animals in each run.

The volume of water or Yerba mate infusion consumed was recorded daily, while food consumption and individual body weight were recorded weekly. Experimental design followed the “Ethical principles and guidelines for experimental animals” (3rd. Edition, 2005) by the Swiss Academy of Medical Sciences (http://www.aaalac.org, accessed on 22 May 2022). Handling of animals as well as experimental protocols were evaluated and approved by the Animal Welfare Committee (CICUAL) of the La Plata School of Medicine, UNLP (T06-01-2022).

### 4.4. Chemical Analysis of Yerba Mate Aqueous Extract

Principal constituents of the aqueous extract of Yerba mate were identified using high-performance liquid chromatography coupled with UV-Visible photodiode array (Agilent 1100 Series); results were processed in ChemStation. Elution conditions for phyto-markers were as previously described [4] on a Phenomenex^®^ C18 column (250 × 4.6 mm id, 5 μm). Solvent A (H_2_O + 0.2% Acetic Acid) and B (methanol + 0.2% Acetic Acid) were mixed in gradient mode as follows: 0 min 90% A, 0–5 min 80% A, 5–45 min 50% A, 45–55 min 20% A; flow rate 0.8 mL/min. Injection volume, column temperature, and UV wavelength were set at 80 μL, 30 °C and 254 nm, respectively. Phytochemical standards and solvents were either from Sigma-Aldrich (Dorset, UK) or our in-house phytochemical library.

All standards were prepared as 1 mg/1 mL (70% Acetonitrile and 30% water) except rutin, which was prepared in 1 mg/10 mL (90% water and 10% acetonitrile). A mother solution with a high concentration of 0.1111 mg/mL for each standard was then serially diluted to 0.0555, 0.0277, 0.011, and 0.0055 mg/mL The plant material (1 g) was extracted with 5 mL of distilled water at 80 °C, left to cool down, and then filtered (Whatman paper) immediately after adding an equal amount of methanol as a preservative to avoid microbial growth during analyses.

### 4.5. Glucose Tolerance Test

The glucose tolerance test (GTT) was run on 12h-fasted animals. Glucose (1.5 g/kg in saline solution) was administrated intraperitoneally, and blood samples were obtained from the retro-orbital plexus under ketamine and midazolam anesthesia (80 mg/kg and 5 mg/kg body weight, respectively), at 0, 15, 30, 60, 90 and 120 min after glucose challenge. Glucose concentration was measured by triplicate in each sample with test strips (Accu-Chek Performa, Roche, Manheim, Germany). Results were expressed as the area under the glucose curve (AUC) in mmol/L/min.

### 4.6. Biochemical Analyses of Plasma

After the three weeks of treatment, blood samples from 6h-fasted animals were collected from the retro-orbital plexus under light halothane anesthesia to measure plasma levels of glucose, triglyceride, insulin, and lipid peroxidation measured as thiobarbituric acid reactive substances (TBARS), and ferric reducing ability of plasma (FRAP). Glycemia determination was performed with commercial test strips (Accu-Chek Performa Nano System, Roche Diagnostics. Mannheim, Germany) and triglyceride levels were determined using a commercial enzymatic colorimetric assay (Wiener Lab, Buenos Aires, Argentina). TBARS were assayed by fluorometric method (Yagi method). Data obtained were expressed as pmol of malondialdehyde (MDA) per mg of plasma protein.

FRAP levels were determined by a colorimetric method (FRAP assay) and expressed as µmol Fe2+/L. Circulating insulin levels were assayed by RIA (antibody against rat insulin) from Sigma Chemical Co., rat insulin standard from Novo Nordisk Pharma, Argentina, and highly purified porcine insulin ^125^I. Insulin resistance (IR) was evaluated by homeostasis model assessment-IR (HOMA-IR) according to the formula [insulin (μU/L) × glucose (mmol/L)]/22.5. β-cell function was quantified by HOMA-β [insulin (μU/L) × 20/glucose (mmol/L)] − 3.5. Since these indexes have been clinically validated in humans but not in rodents, we here compare the values of treatment groups to the values of the control group instead of using a cut-off threshold value.

### 4.7. Glucose Stimulated Insulin Secretion

To elucidate the effect of dietary intervention with Yerba mate on ex vivo insulin secretion, the whole pancreas from each experimental animal was removed immediately after euthanasia and collagenase digestion was employed to isolate pancreatic islets [39]. Ten groups of five isolated islets from each condition groups were incubated for 1 h at 37 °C in 0.6 mL Krebs–Ringer bicarbonate buffer (KRB), pH 7.4, gassed with a mixture of CO_2_/O_2_ (5/95%) with 1% (*w*/*v*) BSA and either 3.3 or 16.7 mM glucose. After that, aliquots from the medium of each experimental condition were taken in order to measure insulin levels by RIA as described above. Insulin secreted into the incubation medium was expressed as ng of insulin/islet/hour.

To study the direct effect of Yerba mate and some of its main phenolic components, namely chlorogenic acid, rutin, caffeic acid and quercetin, 10 groups of five islets from untreated normal rats were incubated for 1 h at 37 °C in 0.6 mL Krebs–Ringer bicarbonate buffer (KRB), pH 7.4, previously gassed with a mixture of CO_2_/O_2_ (5/95%) with 1% (*w*/*v*) BSA and either 3.3 or 16.7 mM glucose, in the absence (control) or presence of Yerba mate (freeze-dried extract, 10 and 100 µg/mL), or the different phenolic compounds mentioned above at 0.1 and 1 µM. As mentioned previously, after 1 h incubation period, samples from the medium were collected for insulin determination by RIA.

### 4.8. Gene Expression by Real-Time PCR (qPCR)

Total RNA was isolated from islets of Langerhans from both experimental groups using TRIzol Reagent (Gibco-BRL, Rockville, MD, USA). Integrity of the obtained RNA was verified by agarose–formaldehyde gel electrophoresis. Cross-contamination with protein or phenol was checked by measuring the 260:280 nm absorbance ratio. Samples with a ratio of 1.8 or lower were discarded. On the other hand, samples were treated with DNase I (Invitrogen, Waltham, MA, USA) to avoid DNA contamination. Reverse transcription-PCR was conducted with SuperScript III Reverse Transcriptase (Invitrogen), oligo-dT and 1 μg of total RNA as template. qPCRs were run in triplicate using FastStart SYBR Green Master mix (Roche) in the iCycler 5 (BioRad) employing 40 cycles (denaturation at 95 °C for 30 s, annealing at 60 °C for 30 s and extension at 72 °C for 30 s). Every run was followed by a melting curve from 55 °C to 90 °C for checking the presence of only one amplicon. Sequences of oligonucleotide primers (Invitrogen) used in these experiments are shown in Table 2. Quantified values were normalized using β-actin as a housekeeping gene employing individual efficiency calculated with a standard curve for each gene.

### 4.9. Statistical Data Analysis

For the statistical analysis of experimental data, the SPSS program (15.0 version, SPSS, Inc, 25 Chicago, IL, USA) was employed. Student’s *t*-test was applied for independent samples with normal distribution, while ANOVA followed by the Tamhane test for similar variance samples were applied to evaluate significance in glucose-stimulated insulin release in the presence of phenolic compounds of Yerba mate. Results are expressed as mean ± SEM. Differences between groups were considered significant when *p* values were <0.05.

## Figures and Tables

**Figure 1 plants-12-02620-f001:**
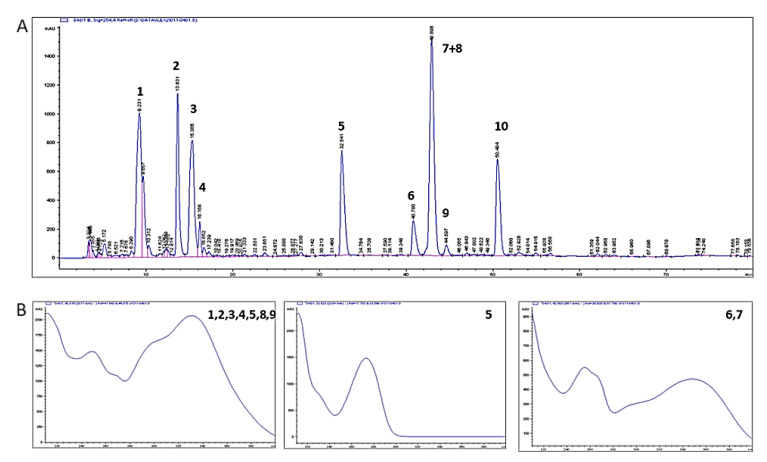
HPLC-DAD analysis of the aqueous extract of Ilex paraguayensis administered to experimental animals. (**A**) Chromatogram (254 nm); (**B**) UV spectra of the labelled peaks.

**Figure 2 plants-12-02620-f002:**
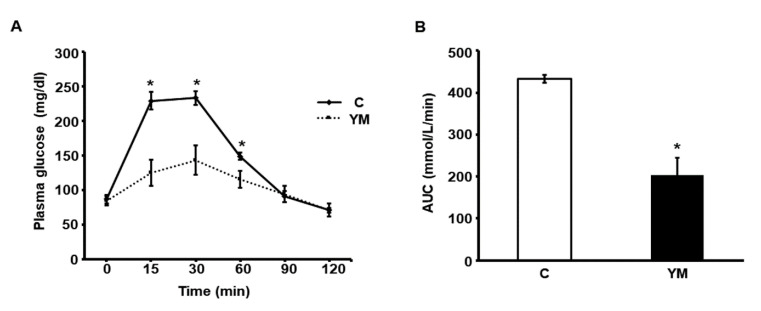
Glucose tolerance test. (**A**) Plasma glucose values (mg/dL) measured in C (solid line) and YM rats (dotted line) at 0, 15, 30, 60, 90 and 120 min after glucose load. (**B**) Area under the glucose curve (AUC) expressed as mmol/L/min, during glucose tolerance test in C (white bars) and YM (black bars) animals. Values are mean ± SEM (n = 6 per group) * *p* < 0.05.

**Figure 3 plants-12-02620-f003:**
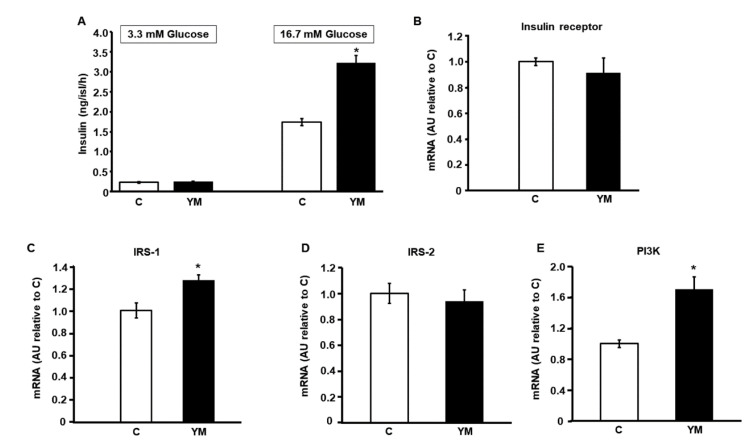
Insulin secretion in response to different glucose concentration and mRNA levels of intracellular mediators of insulin signaling pathway. (**A**) Insulin secretion at 3.3 and 16.7 mM glucose by islets isolated from C (white bars) and YM (black bars) rats. Values were expressed as ng of insulin per islet/h. Bars represent means ± SEM from three independent experiments. * *p* < 0.05. (**B**–**E**) mRNA relative expression (RT qPCR) of insulin receptor (**B**), IRS-1 (**C**), IRS-2 (**D**) and p110 catalytic subunit of PI3K (**E**) in islets isolated from C (white bars) and YM (black bars) rats. β-actin was used as housekeeping gene. Values were expressed as arbitrary units (AU) compared to mRNA level determined in C islets. Each bar represents mean ± SEM from three independent experiments. * *p* < 0.05 vs. C.

**Figure 4 plants-12-02620-f004:**
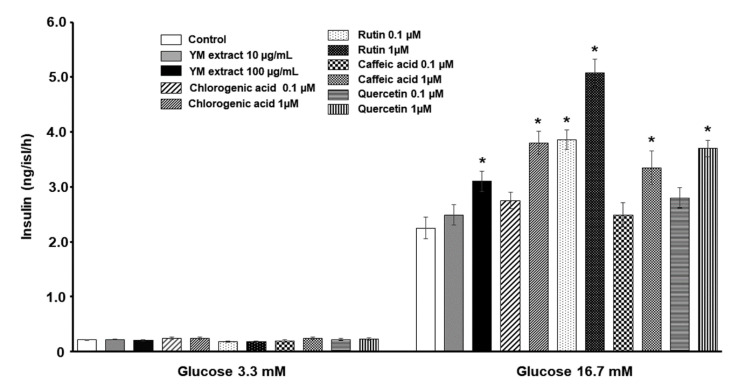
In vitro glucose-stimulated insulin release in presence of phenolic compounds of Yerba mate. Bars represent in vitro effect of the YM freeze-dried extract and phenolic compounds on the insulin release of untreated isolated rat pancreatic islets in the presence of glucose (3.3 mM or 16.7 mM). Each bar represents mean ± SEM from three independent experiments. * *p* < 0.05.

**Table 1 plants-12-02620-t001:** Plasma levels of different metabolic and endocrine parameters.

Plasma Parameters	C	YM
Glucose (mg/dL)	115.14 ± 2.35	108.62 ± 3.43
Triglyceride (mg/dL)	100 ± 7.5	87 ± 8.4
Insulin (ng/mL)	0.39 ± 0.03	0.44 ± 0.08
FRAP (µmol Fe^2+/^L)	255 ± 7.5	412 ± 17 *
TBARS (pmol/mg prot)	598 ± 21	341 ± 33 *
HOMA-IR	2.63 ± 0.27	2.77 ± 0.48
HOMA-β	11.59 ± 1.85	15.86 ± 6.77

Results are shown as mean ± SEM. n = 18 animals per group. * *p* < 0.05.

**Table 2 plants-12-02620-t002:** Commercial oligonucleotide primers used in the study.

Gene	GeneBank	Sequences
Insulin receptor	NM_017071	Fw 5′-ATATTGACCCGCCCCAGAGG-3′Rv 5′-TAGGTCCGGCGTTCATCAGA-3′
IRS-1	NM_012969	Fw 5′-TGTGCCAAGCAACAAGAAAG-3′Rv 5′-ACGGTTTCAGAGCAGAGGAA-3′
IRS-2	NM_001168633.1	Fw 5′-CTACCCACTGAGCCCAAGAG-3′Rv 5′-CCAGGGATGAAGCAGGACTA-3′
PI3K	NM_053481	Fw 5′-GGTTGTTGTTGCCCCAGAC-3′Rv 5′-GGTTGTTGTTGCCCCAGAC-3′
TNF-α	NM_0.126675.3	Fw 5′-GGCATGGATCTCAAAGACAACC-3′Rv 5′-CAAATCGGCTGACGGTGTG- 3′
PAI-1	NM_012620.1	Fw 5′-CCACGGTGAAGCAGGTGGACT-3′Rv 5′-TGCTGGCCTCTAAGAAGGGG- 3′
β-actin	NM_031144.3	Fw 5′-AGAGGGAAATCGTGCGTGAC-3′Rv 5′-CGATAGTGATGACCTGACCGT-3′

Fw: forward primer; Rv: reverse primer.

## Data Availability

Not applicable.

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
