# Peer review of "Dietary Supplementation with Yerba Mate (Ilex paraguariensis) Infusion Increases IRS-1 and PI3K mRNA Levels and Enhances Insulin Sensitivity and Secretion in Rat Pancreatic Islets"

_plants, 2023, doi:10.3390/plants12142620_

Round 1
Reviewer 1 Report
The manuscript entitled “Dietary supplementation with Yerba mate (Ilex paraguariensis infusion) increases IRS-1 and PI3K mRNA levels and enhances insulin sensitivity and secretion in rat pancreatic islets” is an interesting contribution about the effect of Ilex paraguariensis infusion on the several biomarkers in rat pancreatic islets. The introduction is well written, objectives are clear, methodology is reproducible and the conclusions are supported by data, thus, in my opinion, only minor details are needed previous to accept the manuscript
Comments
Introduction
Please add a paragraph (including cites) about the importance of the pancreatic islets
L203-204 Please re-write: “… thereby providing evidence of the importance of regulating inflammation by using natural products found in certain foods…”
L235-237 Please re-write “… Taken together, these findings suggest that long-term dietary supplementation with Yerba mate may be beneficial for improving diseases characterized by glucose homeostasis alterations such as prediabetes or type 2 diabetes.”
The main concern is about the “long term”, because the use of the experimental set-up ( 3 weeks), how to generalize to long term?
Author Response
1. Please add a paragraph (including cites) about the importance of the pancreatic islets
According to the reviewer suggestion, we have added a paragraph and the corresponding references concerning the importance of pancreatic islets (page 2, lines 70 to 80 and page 10, lines 428 to 433).
2. L203-204 Please re-write: “… thereby providing evidence of the importance of regulating inflammation by using natural products found in certain foods…”
The sentence was modified for the sake of clarity (page 6, lines 216 to 218).
3. L235-237 Please re-write “… Taken together, these findings suggest that long-term dietary supplementation with Yerba mate may be beneficial for improving diseases characterized by glucose homeostasis alterations such as prediabetes or type 2 diabetes.” The main concern is about the “long term”, because the use of the experimental set-up (3 weeks), how to generalize to long term?
We agree with reviewer and in consequence we have eliminated the mention to “long term”. The sentence was modified (page 7, lines 252 to 254).
Reviewer 2 Report
Interesting paper with clearly presented results. My only query was why no quantification of constituents by HPLC of the actual Yerba Mate extracts to get an idea of how much of each specific constituent. To give an indication of the amount consumed in the treatment group or for comparison with concentrations of the reference compounds in the in vitro tests
Author Response
Interesting paper with clearly presented results. My only query was why no quantification of constituents by HPLC of the actual Yerba Mate extracts to get an idea of how much of each specific constituent. To give an indication of the amount consumed in the treatment group or for comparison with concentrations of the reference compounds in the in vitro tests
Thanks for your kind comments and suggestion. We indeed quantified the metabolites in the plant material but after a slightly different method of extraction. Therefore, we did not include the data in our submission fearing it would cause some confusion. However, if you believe that these may help the readers to understand the approximate yields for such metabolites, we can add the following paragraph in Methods: "All standards were prepared as 1mg/1 mL (70% Acetonitrile and 30% water), except Rutin which was prepared in 1mg/10mL (90% water and 10% Acetonitrile). A mother solution with a high concentration of 0.1111 mg/mL for each standard was then serially diluted to 0.0555, 0.0277, 0.011, and 0.0055 mg/mL. The plant material (1g) was extracted with 5 mL of distilled water at 80°C, left to cool down, and then filtered (Whatman paper) immediately after adding an equal amount of methanol as a preservative to avoid microbial growth during analyses" and the following paragraph in Results: “The concentration of the plant metabolites of interest in the water extract (before the addition of alcohol as preservative) was calculated as: chlorogenic acid 1.952 mg/mL (R2=0.993), caffeine 1.836 mg/mL (R2=0.999), caffeic acid 0.092 mg/mL (R2=1.000), rutin 0.192 mg/mL (R2=0.997), and quercetin 0.044 mg/mL (R2=0.999).